# Broadband Waterborne Multiphase Pentamode Metastructure with Simultaneous Wavefront Manipulation and Energy Absorption Capabilities

**DOI:** 10.3390/ma16145051

**Published:** 2023-07-17

**Authors:** Yi An, Han Zou, Aiguo Zhao

**Affiliations:** 1School of Physical and Mathematical Science, Nanjing Tech University, Nanjing 211816, China; anyi@njtech.edu.cn; 2College of Civil Engineering, Nanjing Tech University, Nanjing 211816, China

**Keywords:** metamaterials, pentamode materials, acoustic metastructure, dispersion curve

## Abstract

Acoustic metastructures are artificial structures which can manipulate the wavefront in sub-wavelength dimensions, and previously proposed acoustic metastructures have been mostly realized with single materials. An acoustic metastructure with composite structure is proposed for underwater acoustic stealth considering both wavefront manipulation and sound absorption. The unit cells of the metastructure are composed of a metallic supporting lattice, interconnecting polymer materials and mass balancing columns. With the gradual modulations of equivalent physical properties along the horizontal direction of metastructure, the incident acoustic wave is reflected to other directions. Meanwhile, the polymer material inside the unit cells will dissipate the acoustic wave energy due to inherent damping properties. With the simultaneous modulations of reflected wave direction and scattering acoustic amplitude, significant improvement of the underwater stealth effect is achieved. Compared with single-phase metastructure, the Far-Field Sound Pressure Level (FFSPL) of multiphase metastructure decreases by 4.82 dB within the frequency range of 3 kHz~30 kHz. The linearized mean stress for multiphase metastructure is only 1/3 of that of single-phase metastructure due to it having much thicker struts and much more uniform stress distribution under the same hydrostatic pressure. The proposed composite structure possesses potential applications due to its acceptable thickness (80 mm) and low equivalent density (1100 kg/m^3^).

## 1. Introduction

The acoustic stealth performances of submarines are essential in underwater environments, and are mainly determined by the retro-reflected signals detected by active sonars. Similar to stealth technique for radar detecting, acoustic energy absorptive materials such as anechoic coating are generally adopted to reduce the intensity of retro-reflected signals. Traditional anechoic coating made of rubber or polyurethane substrate with cavities are adopted to absorb the incident detecting wave energy or eliminate the radiated noise of the hull, whose stealth performance tends to be very weak at low frequency and high hydrostatic pressure [1,2]. Thick and heavy anechoic coating is required to achieve strong absorption at low frequency, meaning it is unimplementable for most underwater vehicles.

Local resonant metamaterial can manipulate long-wavelength acoustic waves within the subwavelength scale based on the resonance of resonators [3]. The intensive vibration of the resonators around the resonance frequency would lead to strengthened energy dissipation and intensive acoustic absorption. Inspired by this mechanism and introducing the viscoelasticity of the substrate, strong low-frequency waterborne absorption is realized by Wen et al. with a sub-wavelength sample [4], but the inferiority of this is that the effective frequency bandwidth is very narrow. A lot of studies have been conducted to expand the effective frequency bandwidth by introducing multi-resonators [5,6,7,8,9], metal spiral and inter-connecting structures [10,11,12], air cavities and combinations of them [13,14,15,16,17]. For example, Wang et al. proposed a broadband waterborne absorbing metamaterial with gradient cavity array strengthened by a carbon fiber framework [15]. Zhang et al. proposed a waterborne stealth metastructure with a transverse arrangement of carbon nanotubes to broaden the effective absorption range [16]. Fan et al. also proposed an acoustic absorption–bearing metamaterial consisting of four different subunits and the absorption bandwidth increased by 600% [17]. These new designs expanded the effective bandwidth greatly, but still suffer from many issues such as hydrostatic resistance, manufacturing techniques and cost, density, thickness restrictions, etc.

Acoustic metastructure is a gradient-index artificial structure capable of manipulating acoustic waves in an extraordinary way within compact sizes [18,19,20,21], demonstrating promising applications in many practical scenarios such as noise and vibration insulation [22,23,24,25], underwater acoustic stealth [26,27,28], surveying and imaging [29,30,31]. The acoustic stealth mechanism of the local-resonance method is similar to the air cavity-resonance mechanism, where the incident acoustic energy is absorbed and the intensities of retro-reflected signals are diminished. Instead of acoustic energy absorption, an acoustic metastructure could deflect the incident detecting wave into the other direction through gradual modulation of phase and amplitude, which also reduces the retro-reflected acoustic signal intensities received by the detecting sonar. Thus, the proposal of an acoustic metastructure provides an alternative method to eliminate the retro-reflected sound waves. The acoustic metastructure is composed of many sub-units of gradual varying sizes, where both resonant-based and non-resonant-based configurations are adopted for the design of sub-units. Utilizing the thermal loss of air, simultaneous modulations of wavefront and amplitude can be achieved for an airborne acoustic metastructure [32,33], while for a waterborne metastructure, the effective frequency range of resonance-based designs is very small. Thus, non-resonant designs based on pentamode material (PM) are proposed and adopted for the metastructure design.

Pentamode material (PM) is an artificial solid metamaterial with the merits of broadband efficiency and matched impedance with fluids. For a solid material, none of six eigenvalues of elastic modulus equal zero, while for a pentamode material only one eigenvalue of six is not zero [34]. With arbitrarily tailorable equivalent modulus and density [35], PM is adopted to design many different acoustic devices, especially in an acoustic metastructure. Practically, PM acoustic devices are mainly composed of 2D honeycomb-lattice structures [36,37]. Chen et al. designed a broad PM acoustic cloaking with titanium alloy substrate, which is effective in 9 kHz~15 kHz [38]. Zhao et al. conceived a PM device based on titanium alloy (Ti-6Al-4V) mimicking the acoustic properties of water in 3 kHz~30 kHz [39,40]. Su et al. designed an underwater pentamode focusing lens, whose focusing effects were experimentally verified in 20~40 kHz [41]. Chen et al. proposed a high-transmission metastructure which could convert cylindrical waves to plane waves in 15~23 kHz effectively [42]. Sun et al. designed waterborne acoustic carpet with pentamode materials [43]. Zhang et al. designed an underwater acoustic reflective metastructure shifting normal incident waves by 15° within 6~18 kHz, showing great coincidences of experimental measurements with the finite-element simulations [44]. Except for honeycomb-lattice unit cells, several novel PM configurations with square and triangle lattices were proposed by Dong et al. [45], and a reflective metastructure with underwater absorbing capability have been conceived and numerically validated [23,46]. Ren et al. designed a broadband high-efficient gradient lens with square-latticed pentamode metamaterial, which demonstrated excellent subwavelength focusing performance over 5~33 kHz [47].

The above PM devices are realized with single-phase substrate. Zhao et al. proposed a multiphase PM configuration composed of a metallic supporting lattice, interconnecting phase and mass-balancing block [48]. Then, a multiphase PM structure is fabricated and experimentally verified, which demonstrates the robustness of multiphase PM configuration [49,50]. However, in these studies, hard polymer materials (E ≈ 1 GPa) with small damping coefficients are adopted and the damping of the substrates and structures is ignored.

The previously reported waterborne metastructures have mostly been designed with a single material, where the wavefront-manipulating functions and energy-dissipating capabilities could not be achieved at the same time, or it could only be fulfilled at a very narrow frequency band. In this paper, a novel multiphase PM metastructure is put forward, which can simultaneously manipulate acoustic wavefronts and abate the amplitude of reflected wave. Based on the Generalized Snell Law, a directional reflection acoustic metastructure is proposed, whose physical properties are realized with multiphase pentamode unit cells, furtherly. The damping coefficient is introduced through the polymer materials of multiphase unit cell and the corresponding acoustic properties are assessed by COMSOL Multiphysics. For comparison, a single-phase metastructure with the same dimensions is also conceived. The superiorities of the multiphase metastructure are exhibited on aspects of acoustic stealth performance and pressure resistance under high hydrostatic pressure. The results of this paper will promote the practical application of the underwater acoustic metastructure.

## 2. Materials and Methods

In this section, the principle of the Generalized Snell Law is presented. Based on this principle, the physical properties of the abnormal reflection metastructure are derived. Then the geometrical configuration of the single-phase and multiphase pentamode material is conceived. Finally, the method for equivalent physical properties calculation and the Simulated Annealing (SA) algorithm for optimization is described in detail.

### 2.1. Generalized Snell Law (GSL)

When a plane acoustic wave impinges on the metastructure, the relationship between the incident wave and reflected wave follows the Generalized Snell Law (GSL), which is expressed as [18]:(1)$$n_{i}\sin\left(\theta_{r}\right)-n_{i}\sin\left(\theta_{i}\right)=m\frac{\lambda_{i}}{2\pi }\frac{d\Phi \left(x\right)}{dx}$$
where *n*_i_ is the refraction index of the incident medium, *θ*_i_ and *θ*_r_ denotes the incident and reflected angles, *λ*_i_ is the wavelength. *x* is the horizontal coordinate along the metastructure, *Φ*(*x*) denotes the phase variation accumulation and $\frac{d\Phi \left(x\right)}{dx}$ is the phase gradient of reflected wave. *m* is the order of the diffraction peak.

With the Generalized Snell Law, a directional reflection acoustic metastructure could be proposed as presented in Figure 1. The incident angle is set as *θ*_i_ = 0°. The phase accumulation of the wave along the transversal direction of metastructure is [44]:(2)$$\Phi \left(x\right)=\frac{2D}{\lambda \left(x\right)}\times 2\pi =\frac{4\pi Df}{c\left(x\right)}$$
where *D* is the thickness of metastructure, *f* is the frequency and *c* is the velocity.

*K*(*x*) and *ρ*(*x*) is the bulk modulus and density of the metastructure along the transversal direction. The impedance of metastructure should equal to that of background medium:(3)$$Z\left(x\right)=\rho \left(x\right)c\left(x\right)=Z_{0}=\rho_{0}c_{0},c\left(x\right)=\sqrt{\frac{K\left(x\right)}{\rho \left(x\right)}}$$

Combining (1), (2) and (3), the properties of the metastructure are expressed as:(4)$$\rho \left(x\right)=\left(\frac{n_{i}}{2D}\sin\left(\theta_{r}\right)x+Const\right)\rho_{0},\;K\left(x\right)=\frac{Z_{0}}{\rho \left(x\right)}$$
where Const is an integration constant, indicating that the same wavefront modulation function could be obtained with different values of *K* and *ρ*.

### 2.2. Configurations of Unit Cells

The design principle of metastructure is to introduce a designable phase-gradient profile along the surface, by which the behavior of the reflected wave can be manipulated. The design procedure of metastructure is shown in Figure 2. An analytical solution based on Equation (4) is proposed firstly as shown in Figure 2a, where continual physical parameters (the modulus *K*(*x*) and the density *ρ*(*x*)) are derived. Then the parameters along the transversal direction are discretized while the vertical direction is homogeneous (*K*_i_, *ρ*_i_), as shown in Figure 2b. The physical parameters (*K*_i_, *ρ*_i_) have specified combinations, which are generally unavailable from natural materials. Thus, artificial structures with effective modulus and density are desired in acoustic metastructure realizations. Honeycomb configurations with mass-balancing blocks are widely adopted as shown in Figure 3a, where the thickness of struts determines the desired modulus *K* and the additional weights were adopted to adjust the desired density *ρ*. Thus, PM acoustic metastructure designed with single-phase configuration are gradient structures with varying strut thickness and mass-balancing blocks, as shown in Figure 2c, and the typical fabricated sample is shown in Figure 2d. It can be seen that strut thickness changes gradually according to the distribution of the physical parameters.

Single-phase materials are reported for pentamode device designing and fabrication in previous studies. As can be found in most previous works, the metastructure constructed by a series of individually conceived subunits is usually a passive and lossless acoustic device, because the subunits of the metastructure are usually lossless. If unit cells with inherent damping are introduced, a metastructure capable of wave energy loss could also be designed. For waterborne acoustic metastructure, metallic materials with low damping properties are generally adopted as substrate of single-phase configuration, which would lead to a lossless metastructure. Thus, dual-phase and triple-phase pentamode configurations are proposed as presented in Figure 3b,c in this paper. The unit cell contains three sections: the matrix material (metallic alloys in most cases, adjusting the equivalent modulus), connecting material (polymeric materials with high damping coefficient) and balancing material (materials with very high density are usually adopted, adjusting the equivalent mass density).

**Figure 3 materials-16-05051-f003:**
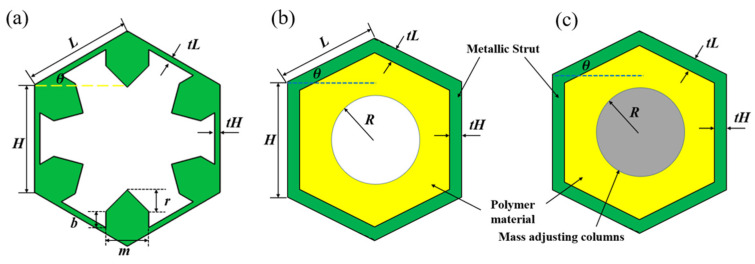
Illustrations of 2D pentamode unit cells. (**a**) Single-phase (SP) pentamode configuration. (**b**) Dual-phase pentamode configuration. (**c**) Triple-phase (TP) pentamode configuration.

The cell geometry of the single-phase pentamode unit (Figure 3a) is characterized by the following parameters:(5)X=[H L θ tH tL m b r]

The two multiphase pentamode configurations presented in Figure 3b,c are depicted by the following parameters:(6)X=[H L θ tH tL R]

### 2.3. Method for Equivalent Physical Properties Calculation

The equivalent density *ρ*_equ_ could be deduced by the areas and densities of each material in the configuration. Homogenization theory is employed to obtain the equivalent modulus *κ*_equ_ firstly, but the studies revealed that the calculated results are only accurate in the long wavelength condition (low frequency range), while the deviation would be huge at higher frequency due to dynamic effect. Thus, dispersion curves are preferred for deriving the equivalent dynamic modulus *κ*_equ_, which is accepted and adopted by most researchers. The wave velocities can be deduced through the slopes of the dispersion curves (*c* = *ω*/*k*) For the acoustic metamaterials, the equivalent compression wave velocity, *c*_L_, and equivalent shear wave velocity, *c*_T_, are given by:(7)$$c_{L}\approx \sqrt{\frac{\kappa_{equ}}{\rho^{equ}}},c_{T}=\sqrt{\frac{G_{equ}}{\rho^{equ}}}$$

From Equation (7), the equivalent bulk modulus and shear modulus are derived. It should be pointed out that the slope will not be a constant especially in the high-frequency region.

Aluminum alloy (*E* = 69 GPa, *v =* 0.33, and *ρ* = 2700 kg/m^3^) is applied as the matrix material, thermoplastic polyurethanes (TPU, *E* = 100 MPa, *v =* 0.4, and *ρ* = 1000 kg/m^3^) is applied as the connecting material, while lead columns (*E* = 16 GPa, *v =* 0.42, and *ρ* = 11,300 kg/m^3^) is chosen as the balancing material. Several parameters are fixed: *H* = *L* = 10 mm, *θ* = 30°, *tH* = *tL*. Typical dispersion curves of the pentamode cells calculated with COMSOL are presented in Figure 4a. The dashed fitting lines correspond to the compression and shear waves in the long-wavelength limit, respectively.

The parametric design of PM microstructure given a specific combination of physical parameters (*K_i_*, *ρ_i_*) is an inverse problem. Due to the numerous combinations of design space, optimization methods are urgent to pursue the optimal design variables. A gradient-free method, Simulated Annealing (SA) technique, is adopted in this study. The SA optimization method is inspired by the annealing process of metals, which was conceived by Metropolis et al. and developed by Kirkpatrick et al. [24,51,52]. A methodology integrating dispersion curve with SA optimization technique is developed by the authors. The optimization process is carried out by COMSOL (v5.4, COMSOL Inc., Stockholm, Sweden) with MATLAB (R2018a, MathWorks Inc., Natick, MA, USA). For any given set of design variables, the equivalent properties are derived through dispersion curve. The SA algorithm integrated with COMSOL was developed and the flow chart is illustrated in Figure 4b, and a detailed description was introduced in a previous work [51].

## 3. Results and Discussions

In this section, the influences of the damping coefficient on the acoustic properties of the ideal metastructure are studied, which are illustrated from the scattering acoustic pressure field map and Far-Field Sound Pressure Level within 3~30 kHz. Then, metastructures with single-phase and multiphase configurations are designed based on the methodology described in Section 2, whose acoustic properties are respectively numerically investigated and quantitatively verified. Finally, the hydrostatic resistance of both metastructures are displayed in terms of stress distribution under the same hydrostatic pressure.

### 3.1. Influences of Damping Coefficient on Acoustic Properties of Ideal Metastructure

In this paper, an aluminum block with *L* = 693.2 mm and *D*_b_ = 70 mm is utilized as the rigid wall, while the thickness of the metastructure is *D* = 80 mm. The angle of the incident wave is set as *θ*_i_ = 0°, while the reflection angle is set as *θ*_r_ = 15°. The properties of the metastructure are derived from Equation (4), where the densities vary from 0.5677*ρ*_0_ to 1.6323*ρ*_0_ and the bulk modulus vary from 1.7616*κ*_0_ to 0.6126*κ*_0_. The average equivalent density of the metastructure is about 1.10*ρ*_0_, which is acceptable in underwater applications. The acoustic properties are investigated by COMSOL Multiphysics. A full 2D model simulation is performed in the “Acoustics-solid, Frequency Domain” module. The isotropic structural damping coefficient *η*_s_ is endowed to the material via the subnode of “Linear Elastic Material”. Perfectly matched layers are set on the outer boundaries of simulation domains to eliminate reflections. The metastructure is set on the upper surface of the aluminum block, while the incident wave impinges on the models from up to down.

The scattering acoustic pressure field maps of the metastructure at 10 kHz, 20 kHz and 30 kHz are shown in Figure 5, respectively. For the aluminum block, strong scattering is observed at all frequencies, and the maximum scattering pressure exits in the incident direction as presented in Figure 5a–c. For the continual metastructure derived from theory, an obvious deflection of scattering wave is observed, as shown in Figure 5d–f, and the reflection angle is about 15°, which is the same as that of theoretical design. Continual material properties are unavailable in practice. Therefore, discretization is a crucial step in practical realization of metastructure. In this study, the continual metastructure is divided into 20 pieces, of which each piece of the material property is identical and the corresponding parameters are presented in Table 1. The density increases from left to right, while bulk modulus varies oppositely. The simulation results of the discretized metastructure are illustrated in Figure 5g–i. No obvious difference is observed between the continual metastructure and discretized metastructure from the acoustic field map, and the successive quantitative calculation will also prove this conclusion. A damping coefficient of 0.1 and 0.2 are introduced in the material properties, and the simulation results are presented in Figure 5j–o. It is seen that the amplitude of the scattering acoustic pressure is reduced obviously, especially at high frequencies.

To assess the acoustic behaviors of the metastructure quantitatively, Far-Field Sound Pressure Level (FFSPL) is calculated in 3~30 kHz. FFSPL is a measure of the scattering pressure of an object, which is usually defined as follows:(8)$$FFSPL=\mathrm{lg}\left|\frac{{p_{r}|}_{r=r_{0}}}{p_{ref}}\right|$$
where $p_{ref}=1\times 10^{-6}\;\mathrm{Pa}$ is the reference pressure, and ${p_{r}|}_{r=r_{0}}$ is the acoustic pressure of the scattering wave at a distance of r0 from the center.

The polar plots of FFSPL for the aluminum block and different metastructures are presented in Figure 6. The maximum FFSPL appears in the azimuth angle of 90° for the aluminum block and at 75° for various metastructures, which agrees well with the results in Figure 5. The polar curve of the continual metastructure is almost identical to that of the discretized metastructure, and the maximum FFSPL is very close to that of the aluminum block. It is revealed that the metastructure can deflect the propagation direction of the reflected wave without dissipating the acoustic energy. The maximum FFSPL of the two metastructures with damping also appears in the azimuth angle of 75°, and the value decreased with the increase of the damping coefficient. These results reveal that the introduction of damping in the metastructure can reduce the amplitude of the reflected acoustic wave without altering the wave manipulation ability.

The above conclusions also suggest a potential application in the acoustic stealth of submarines, where only the retro-reflected acoustic signals are considered. The directional reflecting metastructure with damping could deflect the scattering acoustic wave to the other direction, avoiding detection; meanwhile, the inherent damping properties will also diminish the intensities of the scattering acoustic wave. FFSPL is also adopted to assess the acoustic stealth ability of the metastructure. In order to ensure the robustness of these results, the average value of FFSPL in the range of 90° ± 3° is adopted in the assessment and the results are illustrated in Figure 7. The FFSPL of the metastructures is smaller than that of the aluminum block above 3 kHz (corresponding to a wavelength of 0.5 m, a bit smaller than the length of the metastructure), and a reduction of 10 dB is achieved for most calculated frequencies. The continual metastructure and discretized metastructure without damping yield almost the same results. The average FFSPL of the aluminum block in the frequency range of 3 kHz~30 kHz is 111.79, while that of the continual metastructure and discretized metastructure is 95.97 dB and 97.46 dB, respectively. With the introduction of damping, the FFSPL diminished obviously as the damping coefficient increased. The average FFSPL is 94.15 dB and 91.95 dB, corresponding to a damping coefficient of 0.1 and 0.2. Compared with the discretized metastructure without damping, the introduction of damping resulted in a further reduction of 5.51 dB on the basis of scattering wave reflection. Thus, it is concluded from the simulation results that the introduction of damping will enhance the acoustic stealth performance of the metastructure significantly.

The properties of the metastructure should be justified in terms of the frequency–wavelength aspect ratio with respect to the geometrical dimension of the metastructure. For an acoustic wavefront manipulating device, it is suggested that its length *L* is larger than three times of the wavelength. For the metastructure proposed in this study, *L* = 693.2 mm, so the suggested maximum wavelength is about 231 mm and the corresponding minimum frequency is about 6.5 kHz (*f* = *c/λ*). It is also clearly indicated from Figure 7 of this manuscript that the wave manipulation capability is almost below 5 kHz.

### 3.2. Microstructure Design of Single-Phase and Multiphase Metastructure

With the methodology mentioned in Section 2.3, the geometrical parameters of single-phase unit cells corresponding to 20 pieces of metastructure are obtained and presented in Table 1. The thickness of the struts diminishes gradually, while the additional mass block increases inversely. Four typical dispersion curves of single-phase unit cells are presented in Figure 8. For each piece, there are two cells in the horizontal direction and five cells in the vertical direction. The whole geometrical picture of the single-phase metastructure and five typical unit cells are presented in Figure 9.

In previous studies, only a single-phase substrate was considered and the damping coefficient of the substrate was seldom considered due to very small values (for metallic materials, the damping coefficient is about 0.005). In the design of multiphase configurations, the damping coefficient of the polymer material is too large to be ignored. For thermoplastic polyurethanes (TPU), the damping coefficient is larger than 0.1 at almost all the frequencies, and even larger than 1.0 at high frequencies. Thus, the effect of the damping coefficient should be assessed quantitatively before the design of the multiphase configurations. Two cells corresponding to dual-phase configuration (No. 5) and tri-phase configuration (No. 16) are presented in Figure 10 to demonstrate the effect of the damping coefficient. The damping coefficient of TPU is set as 0.2. It is shown from the dispersion curves that there is only a tiny difference. Quantitative calculation reveals that the consideration of damping will result in an increase of longitudinal wave velocity in the range of 0.1~0.2%, which could be neglected in microstructure design.

Similar to the design of single-phase unit cells, the geometrical parameters of multiphase cells corresponding to 20 pieces of metastructure are also obtained and presented in Table 1. The first 11 cells are dual-phase configurations, while the last 9 cells are tri-phase configurations (the lead column is introduced to achieve the desired density). The final geometrical picture of the multiphase metastructure and five typical unit cells are presented in Figure 11.

### 3.3. Formatting of Mathematical Components

The scattering acoustic pressure field maps of the single-phase and multiphase metastructure at 10 kHz, 20 kHz and 30 kHz are shown in Figure 12. The values of FFSPL at different frequencies are also calculated and presented in Figure 13.

For the single-phase metastructure, the scattering acoustic pressure field maps are similar to those of the discretized metastructure shown in Figure 5. The azimuthal angles of the reflected wave are about 15° at 10 kHz and 20 kHz, but there is a slight variation at 30 kHz and the azimuthal angle turns out to be about 13°. As quantitatively presented in Figure 13a, the FFSPL of the single-phase metastructure is much larger than that of the discretized metastructure beyond 25 kHz, suggesting the influence of chromatic dispersion at high frequencies. This phenomenon can be explained by the dispersion curves presented in Figure 8. It is seen that for unit cells with thick struts (Figure 8a,b), the slope of the longitudinal wave will maintain as a constant over a very broad frequency range, suggesting that the equivalent properties are effective in this frequency range. Meanwhile, for unit cells with thinner struts (Figure 8c,d), the effective frequency range is much smaller, which will have an adverse influence on wave manipulation. Figure 8d indicates that the slope of the longitudinal wave varies gradually beyond 25 kHz, below which the slopes of the longitudinal wave for all the 20 cells are constants. Thus, each unit cell will exhibit the desired properties and the wave control functionality is guaranteed. Beyond 25 kHz, the slopes of longitudinal wave for some unit cells no longer maintain as a constant. Therefore, the cells will exhibit great deviation from desired physical properties and have an adverse effect on wave control functionality. The average FFSPL over the frequency range of 3 kHz~30 kHz is 98.60 dB, which is only 1.14 dB higher than that of the discretized metastructure without damping. The simulation results indicate the effectiveness of the single-phase metastructure and validity of the design method.

For the multiphase metastructure, without considering the damping behavior of TPU, the wave control functionality is well-exhibited below 12 kHz, above which the azimuthal angle of the reflected wave reduces gradually and results in a much larger FFSPL, as indicated by Figure 12d–f and Figure 13b. This phenomenon can also be explained by the dispersion curves. It is seen from Figure 10 that the chromatic dispersion appears at a much lower frequency. The equivalent acoustic properties of the designed cells can be guaranteed at low frequencies, and the azimuthal angle of the reflected wave is in accord with the theoretical design. The introduction of damping will not change the wave control functionality, as indicated by Figure 12g–l, but would abate the amplitude of the scattering acoustic wave efficiently, as indicated by Figure 13c,d. The average FFSPL for the multiphase metastructure, multiphase metastructure with a damping coefficient of 0.1 and multiphase metastructure with a damping coefficient of 0.2 is 99.85 dB, 96.27 dB and 93.78 dB, respectively. Compared with single-phase metastructure, the average FFSPL of the multiphase metastructure with a damping coefficient of 0.2 decreases by 4.82 dB, which demonstrates great advantages in practical application.

### 3.4. Advantages for Withstanding External Pressure

Waterborne metastructures are frequently used in deep-water environments, and it is very important to assess the ability to resist hydrostatic pressure. Taking a water depth of 500 m as an example, the stress field map of the single-phase and multiphase metastructure are presented in Figure 14 (the maximum mesh size is 0.1 mm).The linearized mean stress along the thickness direction of struts is adopted to assess the stress level of unit cells under different hydrostatic pressure, and it is seen that the cells with thinner struts yield higher stress level for both metastructures. The maximum linearized mean stress for the single-phase metastructure is about 251 MPa, while that of the triple-phase MW is only about 74.2 MPa. The results could be also deduced from the parameter design results shown in Table 1, where the thinnest strut of the multiphase metastructure is 0.58 mm, almost twice of that of the single-phase metastructure (0.30 mm). Except for the thicker strut, the presence of polymer materials and lead columns inside the unit cells will also result in a much more uniform and smaller stress distribution. It is seen that the mean stress in the multiphase metastructure is much smaller and therefore much safer in deep-sea environments.

## 4. Conclusions

In conclusion, a novel underwater multiphase metastructure which could manipulate the wavefront and dissipate the acoustic energy simultaneously was proposed. This study suggested that the proposed multiphase metastructure could improve the acoustic stealth ability significantly, which has great potential in underwater applications. The main conclusions drawn are as follows:(1)A multiphase pentamode configuration composed of hexagonal latticed microstructures, polymer materials and mass-balancing lead columns was proposed to realize the desired physical properties. Compared with the single-phase pentamode unit cell which was mostly designed with metallic materials and the damping coefficient was relatively small, significant damping is introduced in the configuration design. Additionally, more degrees of freedom were introduced, which facilitated the designing of the metastructure.(2)An abnormal directional reflection metastructure with a length of 693.2 mm and width of 80 mm was proposed and numerically verified. Both the simulation results of the scattering acoustic pressure field map and the Far-Field Sound Pressure Level (FFSPL) in the frequency range of 3 kHz~30 kHz revealed that the metastructure could reflect the scattering acoustic wave by an azimuth angle of 15°, which was in agreement with the original design. It was also shown that the introducing of material damping will not alter the direction of the scattering acoustic wave, but it could abate the scattering acoustic pressure amplitude obviously.(3)Both multiphase and single-phase metastructures were designed for the same theoretical metastructure. It is revealed that both metastructures demonstrated the abilities of changing the propagation direction of scattering acoustic wave, but the amplitude of the scattering wave could not be abated for the single-phase metastructure due to the lack of damping properties of the single-phase unit cell. Utilizing the damping properties of the polymer materials inside the multiphase unit cells, the multiphase metastructure could abate the amplitude of the scattering acoustic pressure on the basis of reflecting the scattering wave. Quantitative calculations reveal that the average Far-Field Sound Pressure Level for single-phase metastructure decreased by 13.19 dB compared to the aluminum block within the frequency range of 3 kHz~30 kHz, while that of the multiphase metastructure decreased by 4.82 dB compared to the single-phase metastructure.(4)The pressure resistance capabilities of both metastructures were studied and compared. It was illustrated that the linearized mean stress for the multiphase metastructure is only about 1/3 of that of the single-phase metastructure under the same hydrostatic pressure, which suggested that the metastructure designed with a multiphase configuration could withstand three times the hydrostatic pressure than the one designed with a single-phase unit cell.

## Figures and Tables

**Figure 1 materials-16-05051-f001:**
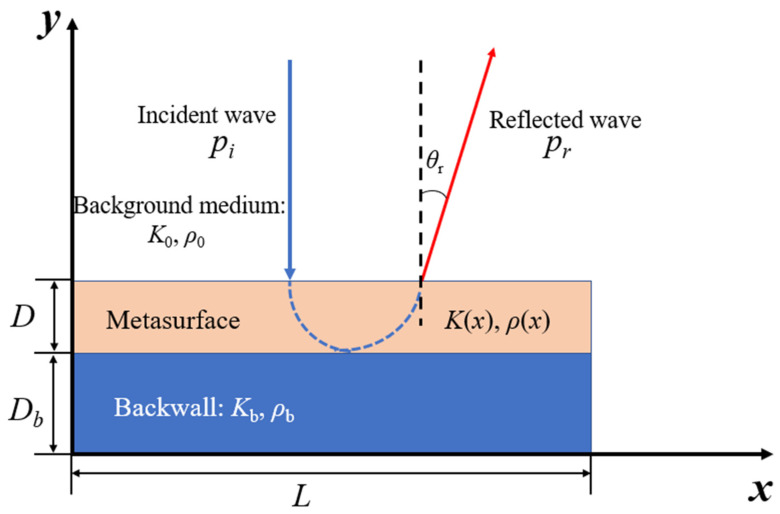
Schematics of normal incident wave and directional reflected wave under the manipulation of a metastructure and backwall.

**Figure 2 materials-16-05051-f002:**
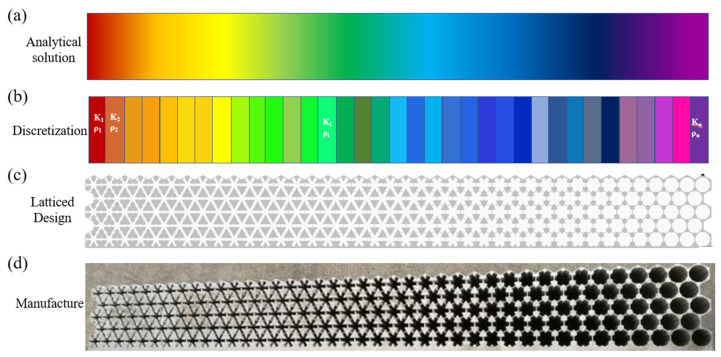
General design procedure for pentamode metastructure devices. (**a**) Continual physical parameters results obtained from analytical solution. (**b**) Discretized physical parameters. (**c**) Unit cell design and the construction of gradient latticed device. (**d**) Fabrication of latticed pentamode device.

**Figure 4 materials-16-05051-f004:**
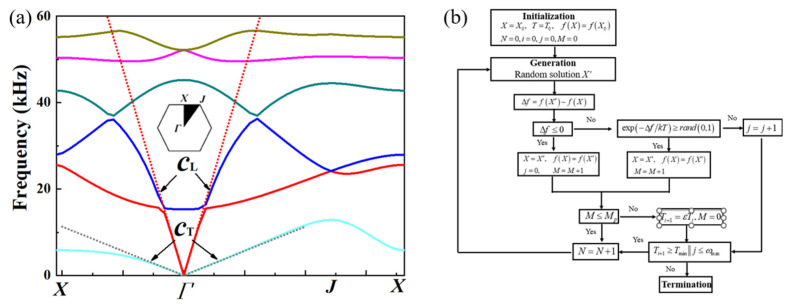
(**a**) Typical dispersion curve for acoustic metamaterials. The solid lines are dispersion curves corresponding to different vibrating modes, and dashed lines are the fitting curve of the first two vibrating modes in the long wavelength regimes (**b**) The flow chart of SA optimization [51].

**Figure 5 materials-16-05051-f005:**
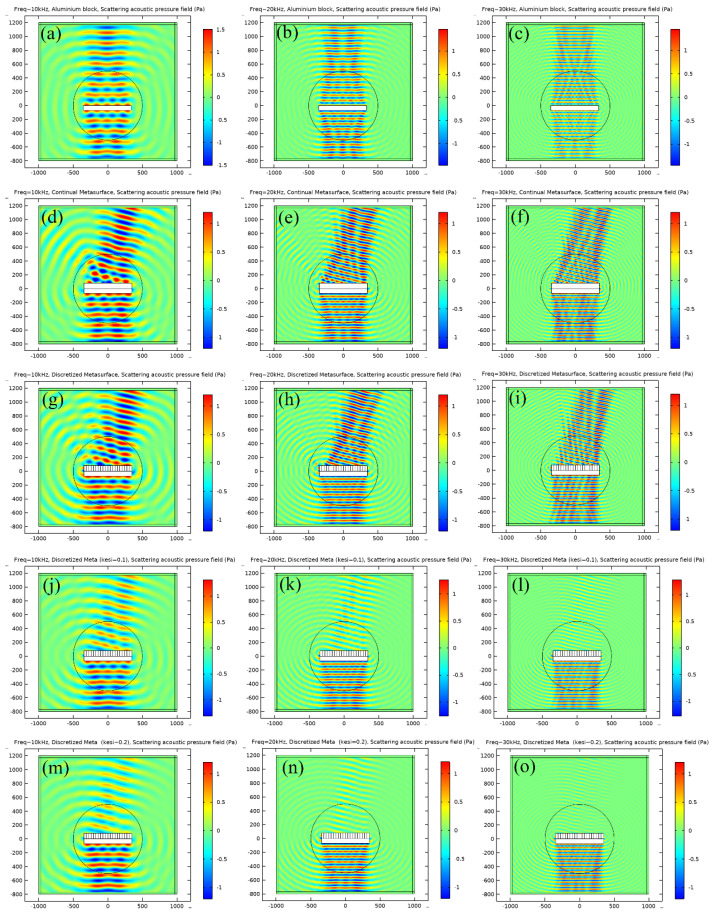
The scattering acoustic pressure field map of the metastructure at 10 kHz, 20 kHz and 30 kHz. (**a**–**c**) Aluminum block. (**d**–**f**) Continual metastructure. (**g**–**i**) Discretized metastructure. (**j**–**l**) Discretized metastructure with damping coefficient of 0.1. (**m**–**o**) Discretized metastructure with damping coefficient of 0.2.

**Figure 6 materials-16-05051-f006:**
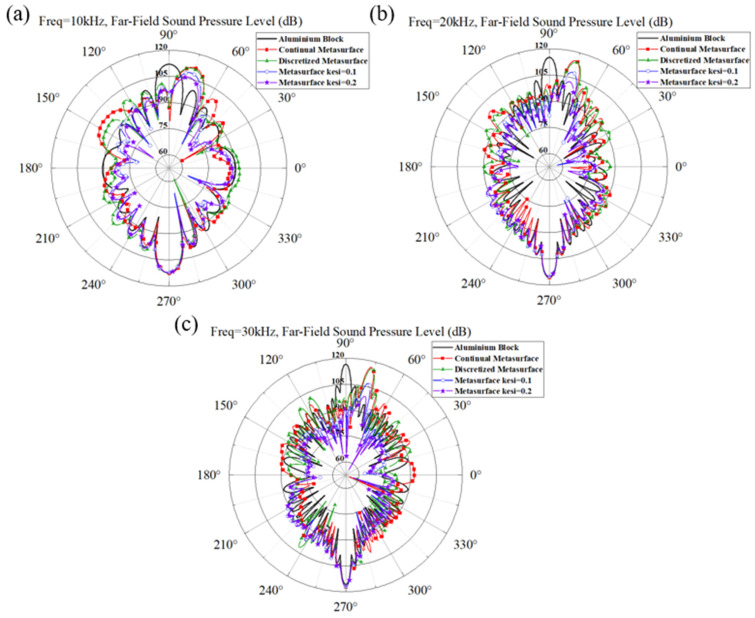
The polar plot of Far-Field Sound Pressure Level (FFSPL) of aluminum block and metastructure at (**a**) 10 kHz, (**b**) 20 kHz and (**c**) 30 kHz.

**Figure 7 materials-16-05051-f007:**
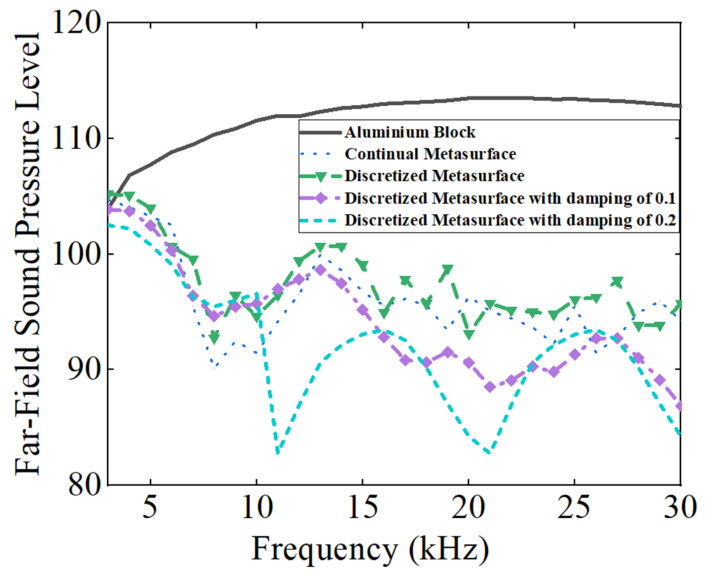
The Far-Field Sound Pressure Level (FFSPL) of different frequencies for aluminum block and metastructure at the incident direction.

**Figure 8 materials-16-05051-f008:**
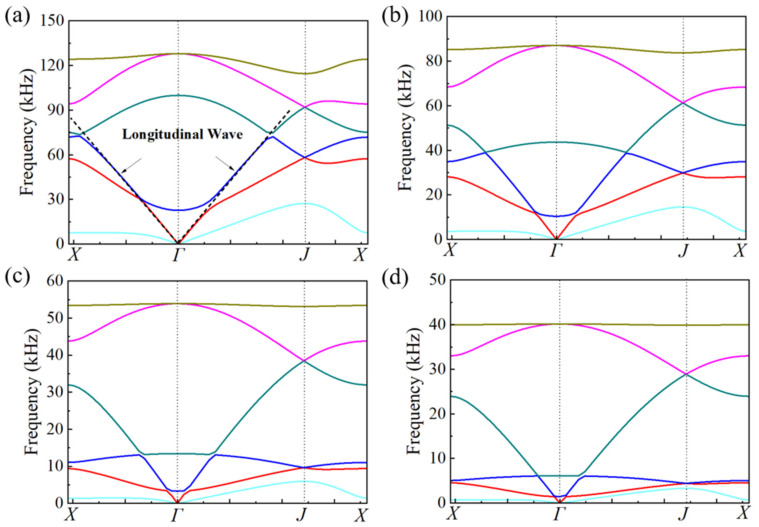
The dispersion curves of four typical single-phase unit cells. (**a**) No.1, (**b**) No.5, (**c**) No.13, (**d**) No.20. The solid lines correspond to the first six vibration modes.

**Figure 9 materials-16-05051-f009:**
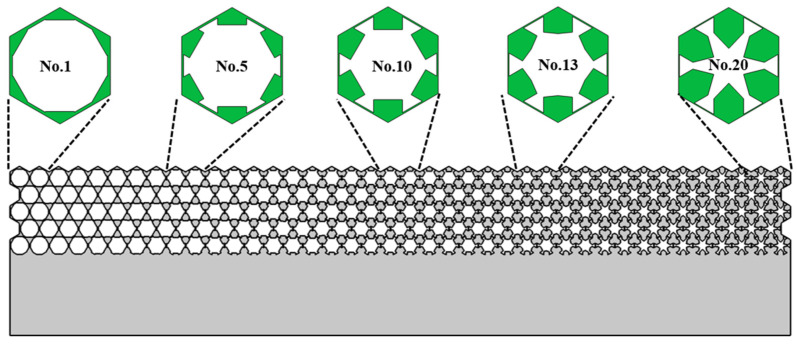
The schematic picture of single-phase metastructure and gradual variation of the unit cells.

**Figure 10 materials-16-05051-f010:**
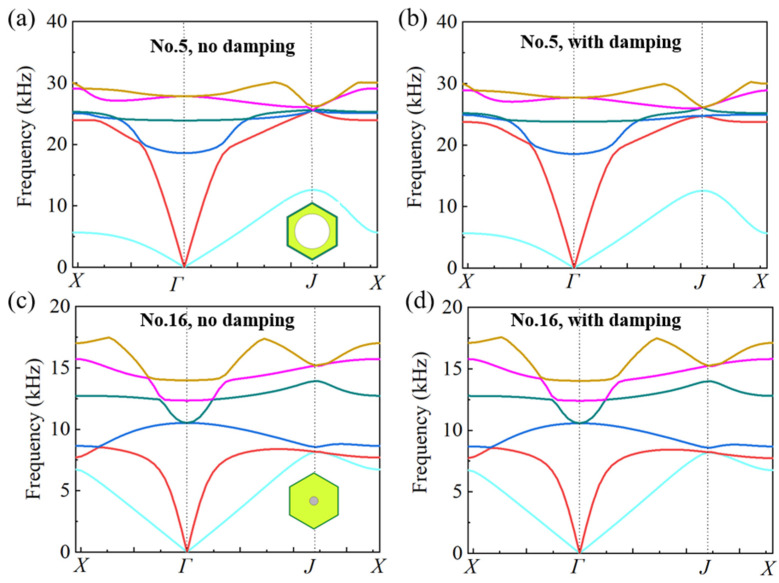
The influence of damping coefficient on dispersion curves of unit cells. (**a**,**b**) correspond to the dispersion curve of No. 5 unit cell where a damping coefficient (0.2) of TPU substrate is considered in (**b**). (**c**,**d**) correspond to the dispersion curve of No. 16 unit cell where a damping coefficient (0.2) of TPU substrate is considered in (**d**). The solid lines correspond to the first six vibration modes.

**Figure 11 materials-16-05051-f011:**
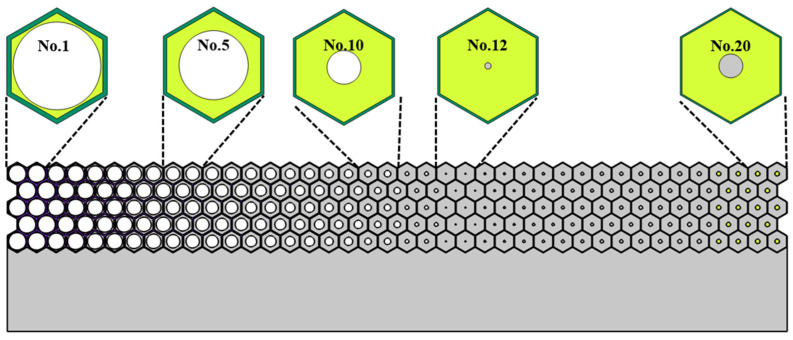
The schematic picture of multiphase metastructure and gradual variation of the unit cells.

**Figure 12 materials-16-05051-f012:**
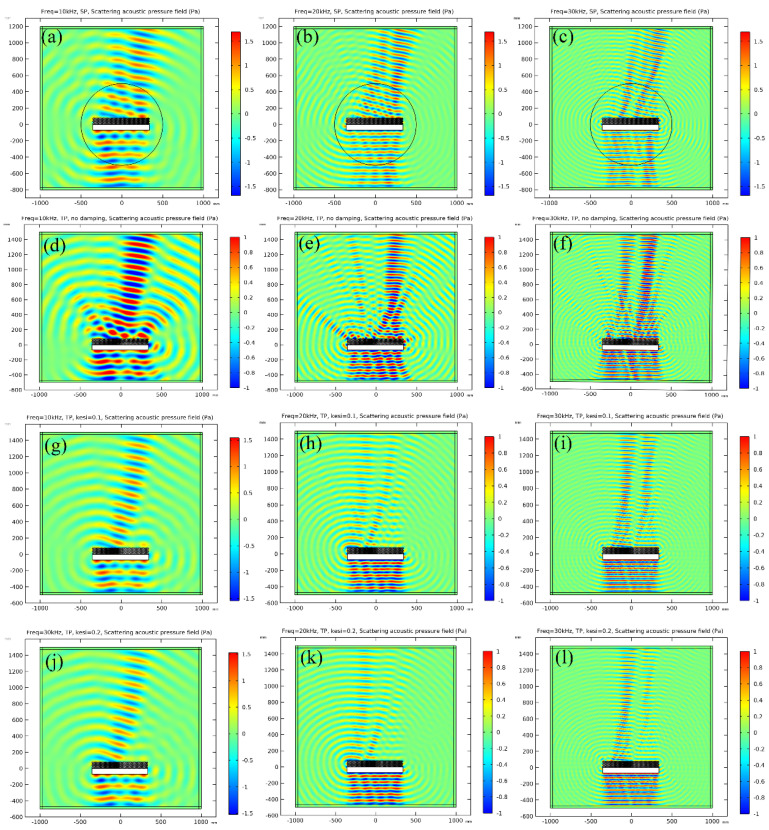
The scattering acoustic pressure field map of the metastructure at 10 kHz, 20 kHz and 30 kHz. (**a**–**c**): Single-phase metastructure. (**d**–**f**): Multiphase metastructure without damping. (**g**–**i**): Multiphase metastructure with damping coefficient of 0.1. (**j**–**l**): Multiphase metastructure with damping coefficient of 0.2.

**Figure 13 materials-16-05051-f013:**
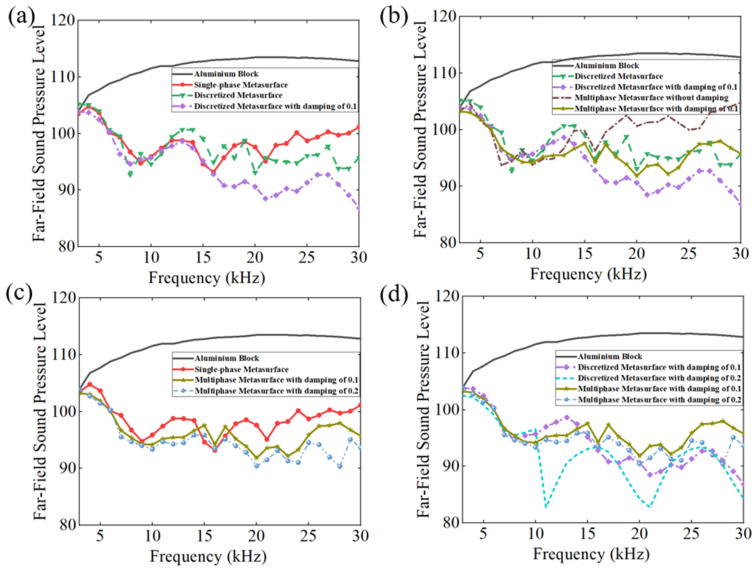
The Far-Field Sound Pressure Level of different frequencies at the incident angle. (**a**) Comparison of FFSPL for single-phase metastructure with theoretical results. (**b**) Comparison of FFSPL for multiphase metastructure with theoretical results. (**c**) Comparison of FFSPL between multiphase metastructure and single-phase metastructure. (**d**) Difference of multiphase metastructure and discretized metastructure with damping.

**Figure 14 materials-16-05051-f014:**
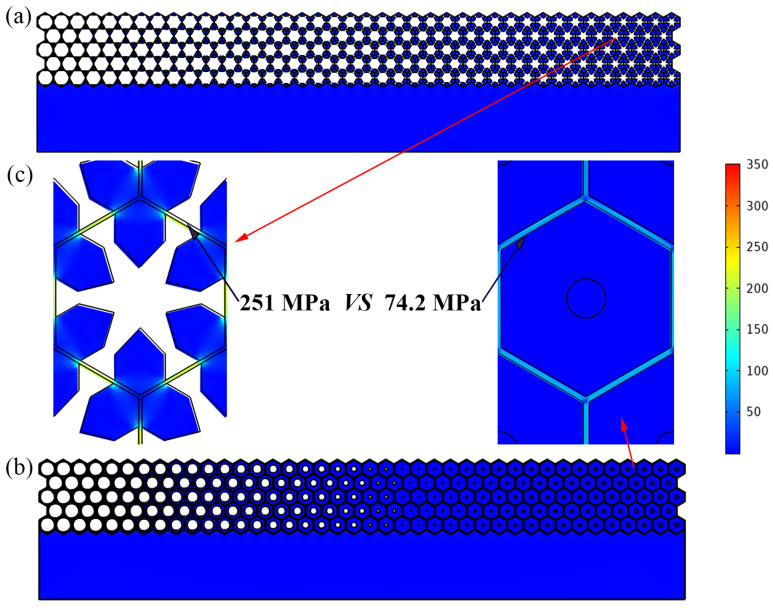
The stress distribution of the proposed metastructure at a hydrostatic pressure of 5 MPa (corresponding to a water depth of 500 m). (**a**) Single-phase metastructure. (**b**) Multiphase metastructure. (**c**) Enlargement for the most dangerous part of the latticed metastructure structure and compassion of the linearized mean stress (251 MPa for single-phase metastructure and 74.2 MPa for multiphase metastructure).

**Table 1 materials-16-05051-t001:** Equivalent properties of the discretized metastructure and corresponding microstructure parameters of the single-phase and multiphase configurations.

Cell No.	Equivalent Properties			Single-Phase				Multiphase	
	X Coordinate (m)	Density (*ρ*_0_)	Modulus (*κ*_0_)	*t* (mm)	*b* (mm)	*m* (mm)	*r* (mm)	*t’* (mm)	*R* (mm)
1	−0.3291	0.5677	1.7616	1.000	2.5	0.13	0	1.55	7.75
2	−0.2944	0.6237	1.6033	0.900	2.5	0.40	0	1.45	7.30
3	−0.2598	0.6797	1.4711	0.820	2.5	0.66	0	1.35	6.90
4	−0.2252	0.7358	1.3591	0.760	2.5	0.89	0	1.25	6.45
5	−0.1905	0.7918	1.2629	0.680	2.5	1.15	0	1.17	5.95
6	−0.1559	0.8478	1.1795	0.630	2.5	1.38	0	1.10	5.50
7	−0.1212	0.9039	1.1063	0.545	2.5	1.64	0	1.03	4.90
8	−0.0866	0.9599	1.0418	0.530	2.5	1.84	0	0.98	4.35
9	−0.0520	1.0159	0.9843	0.515	2.5	2.03	0	0.93	3.70
10	−0.0173	1.0720	0.9329	0.490	2.5	2.23	0	0.88	2.90
11	0.0173	1.1280	0.8865	0.465	2.5	2.44	0	0.82	1.70
12	0.0520	1.1841	0.8446	0.445	2.5	2.64	0	0.77	0.55
13	0.0866	1.2401	0.8064	0.425	2.5	2.70	0.27	0.74	0.90
14	0.1212	1.2961	0.7715	0.405	2.5	2.70	0.67	0.71	1.13
15	0.1559	1.3522	0.7396	0.380	2.5	2.70	1.08	0.68	1.33
16	0.1905	1.4082	0.7101	0.360	2.5	2.70	1.48	0.66	1.50
17	0.2252	1.4642	0.6830	0.350	2.5	2.70	1.85	0.64	1.65
18	0.2598	1.5203	0.6578	0.340	2.5	2.70	2.23	0.62	1.80
19	0.2944	1.5763	0.6344	0.330	2.5	2.70	2.62	0.60	1.92
20	0.3291	1.6323	0.6126	0.300	2.58	2.70	2.68	0.58	2.04

## Data Availability

The data that support the findings of this study are available from the corresponding author upon reasonable request.

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
