# Peer review of "Broadband Waterborne Multiphase Pentamode Metastructure with Simultaneous Wavefront Manipulation and Energy Absorption Capabilities"

_materials, 2023, doi:10.3390/ma16145051_

Round 1

Reviewer 1 Report

This study presents the development of a broadband waterborne multiphase metasurface that can manipulate wavefronts and absorb sound energy simultaneously. The technology developed here has the potential to significantly improve underwater stealth effect. In general, the structure of the article is well organized. The results and analyses are both interesting and valuable. I believe that the topic of the manuscript can be of large interest in material science. However, in my opinion there are some points that need to be improved.

-       I think the originality of the work should be emphasized in the introduction.

-       There are typos in the manuscript. Please go through it again carefully. For instance: On page 6, line 239, figure 4d is written instead of figure 4b.

Author Response

The detailed response to reviewer can be checked in the attached file “Response to reviewers' comments”.

Reviewer 2 Report

I appreciate that authors have analysed numerically the acoustic metasurface and my concerns are-

1.      The paper's Title, abstract and keywords slightly differ from the book chapter (chapter-3, Metamaterial Design and Additive Manufacturing) co-authored by the corresponding author. Therefore, rewrite or update the title, abstract and relevant sections as its impacting the novelty of the manuscript.

2.      Metasurface properties supported by Generalized Snell’s Law, as stated by the authors, need to be justified in terms of Frequency-wavelength aspect ratio with respect to the geometrical dimension of the metasurface.

3.      Apart from design variation of the metasurface as previously published articles from similar author and relevant cited papers, the authors need to highlight the novel contribution with respect to the practical applications.

4.      Mathematical analysis of the metasurface should be supported by physical fabrication and measured results to understand the FFSPL.

5.      Need to address the grammatical and spelling mistakes, improvement of sentence structure with editorial proof reading.

Improvement of sentence structure, grammatical issues is required. 

Author Response

(The authors gave the same response as above.)

Reviewer 3 Report

This paper proposes a metasurface with gradually varying unit-cell for plane-wave absorption and beam manipulation. The introductory analysis is complete and the literature review is very good. Moreover, the paper is well-structured and most of the features are well-established. However, the major issue is the contribution of this paper. In particular:

1) The authors introduce a so-called multi-phase unit cell that includes a polymer material. The main advantage of this material is the possibility to introduce a damping coefficient since the performance without damping is worst compared to the single-phase unit cell. The main question is the following: why not introduce the polymer material to the unit cell of Fig 3a? With this feature, the damping coefficient could, also work with significantly increased bandwidth.

2) The authors use different values of the damping coefficient. It is, also, mentioned that the damping coefficient can be larger than one at high frequencies. As a consequence, how is the 0.1 or 0.2 value selected? Is there any proper reference?

3) The results are compared to the single-phase unit-cell and the theoretical metasurface parameters. The comparison must be extended with results from other state-of-the-art works, such as the cited ones (the better ones from them).

Moreover, some additional comments:

4) How are the equations (1), (2) are derived? A proper citation is required

5) The analysis is conducted in the 2D domain. In a realistic 3D case the proposed device is not a metasurface but a metamaterial since the unit cells extend in 3D. Therefore, the metasurface indication is not completely correct.

6) The last subsection (3.4) provides very limited information. Initially, the colors in Figure 14 are badly linearized since it's almost everywhere blue (the minimum value). Moreover, what are the conclusions from the hydrostatic pressure analysis?

7) The language manipulation is not good. Comprehensive grammar corrections are required.

Extensive editing is required

Author Response

(The authors gave the same response as above.)

Round 2

Reviewer 2 Report

The authors have addressed the relevant concerns. The manuscript can be accepted. 

Author Response

Thanks very much for the reviewer's valuable comments for improving the quality of  the manuscirpt.

Reviewer 3 Report

The authors addressed the minor comments, but the work for the major issues is limited. In particular:

1) The authors claim that the design of the unit cell in Fig. 3a is not convenient, but there is not any clarification for this. Moreover, the analysis in the paper is purely numerical; thus, the simulations of the unit cell in Fig. 3a with polymer material (also with damping coefficient) are totally feasible.

2) The comparison with the results of other works can be tricky. However, the authors must find a way to compare in terms of performance in order to highlight the superiority of their work.

Although all the other comments have been addressed sufficiently, these ones are the most significant, and additional effort is required.

Improved

Author Response

The responses are found in the attached file.
